# The Global Search for Liquid Water on Mars from Orbit: Current and Future Perspectives

**DOI:** 10.3390/life10080120

**Published:** 2020-07-24

**Authors:** Roberto Orosei, Chunyu Ding, Wenzhe Fa, Antonios Giannopoulos, Alain Hérique, Wlodek Kofman, Sebastian E. Lauro, Chunlai Li, Elena Pettinelli, Yan Su, Shuguo Xing, Yi Xu

**Affiliations:** 1Istituto di Radioastronomia, Istituto Nazionale di Astrofisica, Via Piero Gobetti 101, 40129 Bologna, Italy; 2School of Atmosphere Sciences, Sun Yat-sen University, 2 Daxue Road, Xiangzhou District, Zhuhai City 519000, China; baci.dingchunyu@gmail.com; 3Institute of Remote Sensing and Geographical Information System, School of Earth and Space Sciences, Peking University, Beijing 100871, China; wzfa@pku.edu.cn; 4School of Engineering, The University of Edinburgh, Alexander Graham Bell Building, Thomas Bayes Road, Edinburgh EH9 3FG, UK; a.giannopoulos@ed.ac.uk; 5Université Grenoble Alpes, CNRS, CNES, IPAG, 38000 Grenoble, France; alain.herique@univ-grenoble-alpes.fr (A.H.); wlodek.kofman@univ-grenoble-alpes.fr (W.K.); 6Centrum Badan Kosmicznych Polskiej Akademii Nauk (CBK PAN), Bartycka 18A, 00-716 Warsaw, Poland; 7Dipartimento di Matematica e Fisica, Università degli Studi Roma Tre, Via della Vasca Navale 84, 00146 Roma, Italy; sebastian.lauro@uniroma3.it (S.E.L.); elena.pettinelli@uniroma3.it (E.P.); 8Key Laboratory of Lunar and Deep Space Exploration, National Astronomical Observatories, Chinese Academy of Sciences, 20A Datun Road, Chaoyang District, Beijing 100101, China; licl@nao.cas.cn (C.L.); suyan@nao.cas.cn (Y.S.); 9University of Chinese Academy of Sciences, No.19(A) Yuquan Road, Shijingshan District, Beijing 100049, China; 10Piesat Information Technology Co., Ltd, Beijing 100195, China; xingsg@bao.ac.cn; 11State Key Laboratory of Lunar and Planetary Sciences, Macau University of Science and Technology, Avenida Wai Long, Taipa, Macau; yixu@must.edu.mo

**Keywords:** habitability, space missions, space technologies

## Abstract

Due to its significance in astrobiology, assessing the amount and state of liquid water present on Mars today has become one of the drivers of its exploration. Subglacial water was identified by the Mars Advanced Radar for Subsurface and Ionosphere Sounding (MARSIS) aboard the European Space Agency spacecraft Mars Express through the analysis of echoes, coming from a depth of about 1.5 km, which were stronger than surface echoes. The cause of this anomalous characteristic is the high relative permittivity of water-bearing materials, resulting in a high reflection coefficient. A determining factor in the occurrence of such strong echoes is the low attenuation of the MARSIS radar pulse in cold water ice, the main constituent of the Martian polar caps. The present analysis clarifies that the conditions causing exceptionally strong subsurface echoes occur solely in the Martian polar caps, and that the detection of subsurface water under a predominantly rocky surface layer using radar sounding will require thorough electromagnetic modeling, complicated by the lack of knowledge of many subsurface physical parameters. Higher-frequency radar sounders such as SHARAD cannot penetrate deep enough to detect basal echoes over the thickest part of the polar caps. Alternative methods such as rover-borne Ground Penetrating Radar and time-domain electromagnetic sounding are not capable of providing global coverage. MARSIS observations over the Martian polar caps have been limited by the need to downlink data before on-board processing, but their number will increase in coming years. The Chinese mission to Mars that is to be launched in 2020, Tianwen-1, will carry a subsurface sounding radar operating at frequencies that are close to those of MARSIS, and the expected signal-to-noise ratio of subsurface detection will likely be sufficient for identifying anomalously bright subsurface reflectors. The search for subsurface water through radar sounding is thus far from being concluded.

## 1. Water Inventory on Mars

Most of the ice present on Mars today is located in the polar regions, which are covered by ice sheets extending for millions of square kilometers and possessing a thickness of thousands of meters. These deposits are constituted by several geologic units differing in origin, composition, and age. The most recent and dynamic ones are the seasonal deposits of CO${}_{2}$ ice, produced by the condensation of the atmosphere and persisting through the Martian winter with a thickness below one meter [1]. The residual ice caps cover only part of the polar ice sheets and consist of high-albedo deposits of water ice with a thickness well below that of the ice sheets themselves [2]. Below them lie the so-called Polar Layered Deposits (PLD), which consist of hundreds of layers of ice mixed with dust in proportions that differ in every layer depending on climatic conditions at the time of deposition. The overall dust content of the Southern PLD (SPLD) is estimated to be at ≈15% by volume [3], while that of the North PLD (NPLD) is lower, at 5% or less over Gemina Lingula [4], and above 6% overall [5]. The SHARAD radar sounder detected deposits of CO${}_{2}$ ice hundreds of meters thick on top of the SPLD [6]. There are older and dustier ice-bearing deposits beneath the PLD, known as the Basal Unit in the North and the Dorsa Argentea Formation in the South. The Basal Unit (BU) is a deposit consisting of water ice and lithic fines, lying stratigraphically beneath the North Polar Layered Deposits. It consists of two geologic units, namely the Rupes Tenuis unit at the bottom, and the Boreum Cavi unit on top, both of Amazonian age. The Boreum Cavi unit appears to consist predominantly of sandy material [2]. The Dorsa Argentea Formation (DAF) is a vast Hesperian-aged unit surrounding and partially underlying the South Polar Layered Deposits. Volcanic activity, debris flows, aeolian deposition, and glacial activity have been proposed as formation mechanisms [2], but the hypothesis that the DAF is the remnant of a large ice sheet [7] seems to be better supported by evidence.

Ice is also present in mid-latitudes landforms such as lineated valley fills and lobate debris aprons, which are thought to be the remains of glaciers from a recent ice age [8]. In addition to that, neutron spectroscopy revealed the widespread occurrence of ground ice outside the polar caps, even at low latitudes [9]. The depth to which such ice extends is unknown, but it is thermodinamically limited by the lower boundary of the cryosphere, which is the volume of the subsurface in which ground ice is stable. The cryosphere extends from a few meters to several kilometers below the surface, depending mainly on the geothermal heat flow from the interior of the planet and the thermal properties of the crust [10]. Lastly, water molecules can be bound to minerals by processes such as alteration, hydration, and serpentinization.

The total volume of ice currently present at Mars has been estimated to be equivalent to a layer of 34 m over the entire surface of the planet [11] (this quantity is usually referred to as Global Equivalent Layer or GEL), 22 m of which are in the polar deposits [12]. The quantity of liquid water present in early Mars has been estimated through various methods, for example the study of geological features is presumed to have been carved by its action. Valley networks are systems of branching valleys found in the most ancient terrains of Mars and resembling fluvial drainage basins, which appear to have formed in the Noachian age approximately between 4.1 and 3.7 Gyr ago. A recent estimate of the total volume of water needed to carve them led to a conservative lower limit of 640 m GEL of water present at the time of their formation [13].

The ratio between normal and deuterated water on Mars has changed over time, because lighter H${}_{2}$O molecules escape form the planet more easily than heavier HDO. This fact has been used to extrapolate the total amount of water lost over the ages. The measured D/H ratio in the current Martian atmosphere is much higher than the one observed in ancient rocks such as the Martian meteorites, confirming the loss of a large quantity of water inferred from observed atmospheric loss rates [14,15]. It has been estimated [16] that the total water present on the surface of Mars 4.5 Gyr ago must have been 6–7 times the quantity existing today.

## 2. The Search for Liquid Water

Liquid water can now be present at the surface of Mars only briefly and under uncommon circumstances because of low temperature and atmospheric pressure. However, there is ample geological [17] and mineralogical [18] evidence that water once flowed on the surface of the planet, whose climate thus had to be very different from the current one, at least for part of its history. Due to its significance in astrobiology, assessing the amount and state of liquid water present on Mars today has become one of the drivers of its exploration.

Evidence of a geologically recent (i.e., less than a few million years) occurrence of liquid water at the surface of Mars was first reported in [19], in which networks of narrow, incised channels called gullies were interpreted as being carved by groundwater seepage and surface runoff. Gullies were initially discovered on steep slopes, mostly on impact crater walls, but were later found also in different settings such as sand dunes [20]. A Martian gully is characterized by an alcove at its head, an incised channel, and a downslope depositional apron. The volume of the apron is lower than the volume of the material that has been removed to form the alcove and channel, thus suggesting that some volatile component was initially part of the material flowing through the gully and was lost after its formation [21]. The mapping of gullies over the Martian surface has shown that they occur in the 30–90${}^{\circ }$ latitude band of both hemispheres, and that their presence is anti-correlated with massive ice deposits. Gullies in the 30${}^{\circ }$–40${}^{\circ }$ latitude range are pole-facing, while those polewards of 40${}^{\circ }$ are predominantly oriented toward the equator. Such a distribution appears to be related to the availability of near-surface ice deposits [22]. The formation of gullies has been explained through different mechanisms, some of which do not require liquid water, such as dry granular flows in the presence of CO${}_{2}$ ice. Terrestrial formation mechanisms that have been considered potential analogs for Martian gullies include pyroclastic flows and dry snow avalanches (as examples of natural dry granular flows), and fluvial flows, debris flows, and slushflows as processes involving the presence of liquid water. As discussed in [23], morphological evidence and laboratory experiments seem to point to liquid-water debris flows resulting from surface melting as the most plausible formation mechanism for gullies.

High-resolution imaging has recently revealed the presence of the so-called recurring slope lineae, which are narrow (a few meters wide), dark streaks occurring on Sun-facing steep slopes close to the equator. Appearing and gradually growing during warm seasons, they fade in cold seasons [24], and have been interpreted as either water flows caused by the melting of ground ice or dry grain flows [25]. Spectrographic analysis of recurring slope lineae has provided no evidence of water, but it has revealed the presence of perchlorated salts [26], which would lower the freezing point of subsurface water brines. Recent observations, however, point to a dry grain flow mechanism at the origin of recurring slope lineae (e.g., [27]).

Evidence for surface liquid water in the current Martian climate is inconclusive, but water could be present underground below the cryosphere. In the early warm Mars, water would naturally percolate into the ground until it reached an impermeable layer, thus forming an aquifer similarly to what happens on Earth. As the mean surface temperature decreased over the ages, a global cryosphere would form, which would effectively seal groundwater in place [11]. There is widespread evidence for groundwater upwelling in the Martian past, requiring the presence of a global groundwater system (e.g., [28,29]). It has been suggested that such system could be replenished by surface water through the basal melting of the polar caps [30], but estimates of lithospheric heat flow for the current epoch are less than one fourth those of Earth, making basal melting unlikely [31].

Recently, evidence for subglacial liquid water beneath the South polar cap has been obtained through orbital radar sounding [32]. Quantitative analysis of radar echoes from an anomalously bright reflector, about 20 km across at a depth of ≈1.5 km, yielding estimates of its relative permittivity and matching that of water-bearing materials. Alternative mechanisms producing strong basal echoes are the presence of a CO${}_{2}$ ice layer at the top or the bottom of the SPLD, or a very low temperature of the H${}_{2}$O ice throughout the SPLD, enhancing basal echo power compared to surface reflections. However, such phenomena either require very specific physical conditions or they do not cause sufficiently strong basal reflections. Thermophysical modeling of the conditions needed to generate liquid water beneath the South polar cap yields estimates of the required lithospheric heat flow exceeding accepted values for Mars. This result seems to imply the presence of a subsurface thermal anomaly for liquid water to be present [33]. Modeling of the subglacial hydraulic potential beneath the South polar cap, based on radar-derived basal topography, provided estimates of the location of subglacial lakes that do not match the bright radar reflector. This finding is consistent with a hydraulically isolated liquid body confined by cold-based ice, rather than with a subglacial lake [34]. In spite of the theoretical difficulties in reconciling the presence of liquid water with the known characteristics of the SPLD, recent observations acquired by MARSIS over the same region, and analyzed using signal processing procedures commonly applied on Earth to discriminate between wet and dry subglacial areas, are in agreement with the earlier detection of subglacial water, and provide evidence for other wet areas in its surroundings, suggesting the presence of a complex hydrologic system [35].

## 3. Radar Sounding and Subsurface Water Detection

Subglacial water was detected by the MARSIS [36] radar sounder aboard the European Space Agency spacecraft Mars Express. Orbital radar sounding is based on the same principle as radioglaciology; a well-established geophysical technique employed since the mid-20th century to probe the interior of ice sheets and glaciers in Antarctica, Greenland, and the Arctic [37]. It is based on the transmission of radar pulses at frequencies in the Medium Frequency (MF, 300 kHz – 3 MHz), High Frequency (HF, 3–30 MHz) and Very High Frequency (VHF, 30–300 MHz) range into the surface, to detect signals reflected from dielectric discontinuities associated with compositional and/or structural changes in the subsurface. Radar sounders have been successfully employed in planetary exploration since the times of the Apollo program (e.g., [36,38,39,40,41]), and they still are the only remote sensing instruments allowing the study of the subsurface of a planet from orbit. In particular, by transmitting a 1 MHz-bandwidth pulse centered at 1.8, 3, 4, or 5 MHz, MARSIS has been able to reveal echoes coming from depths of more than 3.5 km beneath the Martian polar cap [12].

An electromagnetic wave encountering a discontinuity in the medium through which it is propagating is partially reflected, while the remainder is transmitted. In the ideal case of a plane parallel geometry in which the wave is perpendicular to the discontinuity, the partition between reflected and transmitted power is described by the reflectance (or reflectivity, or power reflection coefficient) R, and the transmittance (or transmissivity, or power transmission coefficient) T at normal incidence [42]:(1)$$R={\left|\frac{\sqrt{\varepsilon_{1}}-\sqrt{\varepsilon_{2}}}{\sqrt{\varepsilon_{1}}+\sqrt{\varepsilon_{2}}}\right|}^{2}$$
(2)T=1−R
where $\varepsilon_{1}$ is the complex relative permittivity in the medium from which the electromagnetic wave propagates and $\varepsilon_{2}$ is the same parameter for the medium past the discontinuity. It can be seen from Equation (Equation 1) that the greater the difference between $\varepsilon_{1}$ and $\varepsilon_{2}$, the more energy is backscattered towards the radar transmitting the electromagnetic wave.

The equation describing the amount of power received by a radar illuminating some target is known as the radar equation. A specialized form of this equation for an orbiting radar sounder over a plane surface has been presented in [43]:(3)$$P_{s}=P_{t}\times {\left(\frac{G\lambda }{8\pi H}\right)}^{2}\times R_{s}$$
where $P_{s}$ is the power of the surface echo received by the radar, $P_{t}$ is the power of the transmitted pulse, *G* is the radar antenna gain, λ the pulse wavelength, *H* the spacecraft altitude, and $R_{s}$ the Fresnel reflection coefficient at normal incidence for the surface. The geometric term ${\left(\lambda /\left(8\pi H\right)\right)}^{2}$ represents the geometric loss due to the spherical expansion of the wavefront (both in transmission and after reflection) multiplied by the squared area of the Fresnel circle producing the specular reflection.

The term $R_{s}$ in Equation (Equation 3) implies that part of the radar pulse energy propagates into the subsurface and can be reflected back to the radar in the presence of a subsurface dielectric discontinuity. In this case, the subsurface echo power $P_{ss}$ received by the radar can be computed through the following expression [43]:(4)$$P_{ss}=P_{t}\times {\left(\frac{G\lambda }{8\pi (H+z)}\right)}^{2}\times T_{s}^{2}\times R_{ss}\times \exp\left(-2\pi f\tan\delta \;\tau \right)$$
where *z* is the depth of the subsurface dielectric discontinuity, $T_{s}$ the surface transmission coefficient, $R_{ss}$ the subsurface Fresnel reflection coefficient at normal incidence, *f* the radar frequency, tanδ the loss tangent of the medium between the surface and the subsurface discontinuity, while τ is the time delay between the reception of the surface and subsurface echoes. The loss tangent is the ratio between the imaginary and real parts of the complex relative permittivity, and the term $\exp\left(-2\pi f\tan\delta \tau \right)$ expresses the attenuation of the radar signal because of dielectric losses as it propagates through the subsurface. Depth *z* and subsurface echo delay τ are related through the following expression:(5)$$z=\frac{c\;\tau }{2\sqrt{\varepsilon_{s}^{\prime }}}$$
where *c* is the speed of light in vacuo and $\varepsilon_{s}^{\prime }$ is the real part of the relative permittivity of the medium between the surface and subsurface interface. In the following discussion we will refer to such a medium as the surface layer, while the material below the discontinuity causing the reflection will be called the basal layer. Both layers are considered homogeneous unless stated otherwise. For ease of reference, the real part of the relative permittivity will be called, although somewhat improperly, dielectric constant.

The identification of liquid water in a radar signal is based on its different electromagnetic response compared to ices and other geomaterials. The surface of Mars is constituted predominantly by igneous rocks (e.g., [44]), while its polar caps consist mostly of water ice together with dry ice and dust [2]. Although hydrated minerals have been identified on the planet, they cover only a small fraction of its surface [44]. Table 1 below lists the values of relative permittivity for these materials, together with those of liquid water and brine.

Most materials listed in Table 1 do not exhibit a strong dependence of their relative permittivity on temperature in the range expected for the Martian surface. The loss tangent of water ice, however, increases by orders of magnitude with temperature [47], drastically affecting the attenuation of the radar wave (see Equation (Equation 4)). The loss tangent can be computed as a function of temperature using formulas presented in [47] and is shown in Figure 1 below. It can be seen that cold, pure water ice has a loss tangent that is orders of magnitude below that of other substances, resulting in very little attenuation, and it is thus extremely transparent to radar waves. As ice temperature approaches the melting point, its tanδ becomes higher than that of rock, thus strongly limiting the penetration of the radar signal.

Materials on the surface of Mars can consist of mixtures of different substances, as in the case of the ground ice found outside the polar caps [9], or the Polar Layered Deposits. There are many different models for the effective relative permittivity of a mixture of materials (see e.g., [49] for a discussion), several of which are specialized for particular geometries within the medium. Due to a lack of knowledge about the small-scale structure of Martian materials, past studies have often resorted to the simple and yet widely used Polder-van Santen model. This model has the special property that it treats the inclusions and hosting material symmetrically, i.e., it balances both mixing components with respect to the unknown effective medium, using the volume fraction of each component as a weight. Its formula, as given in [49], is:(6)$$\left(1-v\right)\;\frac{\varepsilon_{h}-\varepsilon_{\mathrm{eff}}}{\varepsilon_{h}+2\varepsilon_{\mathrm{eff}}}+v\;\frac{\varepsilon_{i}-\varepsilon_{\mathrm{eff}}}{\varepsilon_{i}+2\varepsilon_{\mathrm{eff}}}=0$$
where *v* is the volume fraction of inclusions in the mixture, $\varepsilon_{h}$ is the relative permittivity of the host material, $\varepsilon_{i}$ that of the inclusions, and $\varepsilon_{\mathrm{eff}}$ the relative permittivity of the mixture. This formula requires algebric manipulation to obtain an expression for the solution. The result is a quadratic equation with two roots: The correct solution must be greater than 1 and comprised between the relative permittivity values of the host and of the inclusion.

According to Equation (Equation 6), materials such as porous rocks should increase their relative permittivity if their pores are saturated with water and experimental evidence on Earth shows that permittivity values greater than 15 are seldom associated with dry materials [50]. The high relative permittivity of water bearing materials will result in a high reflection coefficient, according to Equation (Equation 1), and indeed the detection of subglacial lakes by means of radar sounding (which in the context of Earth polar studies is called Radio-Echo Sounding, RES) is chiefly based on the detection of an increase in basal echo strength relative to the immediate surroundings (e.g., [51]). This was also the main evidence in identifying subglacial water on Mars [32], but because of the low spatial resolution of MARSIS, it was not possible to corroborate the identification through qualitative information such as bedrock morphology in the radar image, which is an important criterion in terrestrial studies. For this reason, a probabilistic inversion method based on Equations (Equation 3) and (Equation 4) had to be developed to estimate the dielectric constant of the material below the South Polar Layered Deposits of Mars [52], obtaining values above 20 that require the presence of liquid water.

The identification of liquid water on Mars through radar sounding is thus based on the detection of areas of strong subsurface echoes. Indeed, in the case of [32] it was found that echoes coming from below the polar ice sheet at a depth of ≈1.5 km were stronger than surface echoes by several dB, as shown in Figure 2 below.

Subsurface echoes can be stronger than surface echoes because of the higher relative permittivity of water-bearing materials. This can be verified computing the ratio between subsurface and surface echo power by dividing Equation (Equation 4) by Equation (Equation 3):(7)$$\frac{P_{ss}}{P_{s}}=\frac{R_{ss}}{R_{s}}\;T_{s}^{2}\times \exp\left(-2\pi f\tan\delta \;\tau \right)$$
where we neglect the difference between the term (Gλ)/(8πH) in Equation (Equation 3) and the term (Gλ)/(8π(H+z)) in Equation (Equation 4) because MARSIS operates between 250 and 800 km of altitude probing the Martian subsurface down to a few kilometers, and thus z≪H. As mentioned above, this expression assumes that both the surface and subsurface interface are plane and parallel, and thus the effects such as scattering due to surface roughness can be neglected. The Rayleigh roughness criterion is used to determine if a surface can be considered specular:(8)$$∆h<\frac{\lambda }{8\;\cos\theta }$$
where ∆h is the maximum standard deviation of the topographic height for a surface to be considered specular, λ is the wavelength of the incident electromagnetic radiation, and θ is the angle of incidence. For normal incidence and a frequency of a few MHz as in the case of MARSIS, ∆h is of the order of a few tens of meters, whereas the topographic height variation in the area where water was detected is of the order of a few meters over areas of the size of the MARSIS footprint [53]. Although the standard deviation of the topography at the bottom of the polar cap cannot be determined from radar measurements alone, we will assume that it is negligible at least in the area of strong subsurface reflections in which water was identified.

The relative permittivities of surface and subsurface materials have to be defined to compute values of $P_{ss}/P_{s}$ through Equation (Equation 7). The dielectric constant of the surface layer is varied between the lowest value for dry materials reported in Table 1, corresponding to 2.2 for CO${}_{2}$ ice, to a value of 9 matching dense volcanic rocks, to explore the effect of surface layer composition on the strength of subsurface echoes. To simplify the analysis, and as a way to maximize $P_{ss}/P_{s}$ by neglecting dielectric losses, the loss tangent of the surface layer is assumed to be negligible. To compare the most favorable cases for the occurrence of strong subsurface reflections by dry and water-bearing materials, the relative permittivity of the dry bedrock has been assumed to be the highest in the range of values for dry volcanic rocks in Table 1, that is $\varepsilon^{\prime }=9$ and $\tan\delta =10^{-2}$, while the relative permittivity of the liquid water body has been taken to be the highest reported for brines in Table 1, namely $\varepsilon^{\prime }=110$ and tanδ=100. Brines are considered to be more plausible than liquid water as the source of strong basal echoes because temperatures at the base of the SPLD are expected to be well below the freezing point of water (see discussion in [32]). Once the relative permittivities have been defined, the terms $R_{ss}$, $R_{s}$, and $T_{s}$ in Equation (Equation 7) can be computed by means of Equations (Equation 1) and (Equation 2).

It can be seen in Figure 3 that the occurrence of $P_{ss}/P_{s}>1$ in the absence of water-bearing materials is possible only for a dielectric constant below that of water ice, leaving CO${}_{2}$ ice as the main possible constituent of surface material (Table 1). However, CO${}_{2}$ ice is considered to be a minor component of the Southern polar cap of Mars [54], and it has not been detected outside the polar regions. By contrast, in the absence of appreciable attenuation within the surface material, brine would produce strong reflections even if the permittivity of surface material was that of basaltic rocks (Table 1).

Attenuation, as measured by the loss tangent, is the other key factor determining the relative power of surface and subsurface echoes. By setting $P_{ss}/P_{s}=1$ as a limit condition, Equation (Equation 7) can be inverted to determine the values of tanδ that are compatible with the occurrence of strong subsurface echoes. Figure 4 has been produced assuming a basal relative permittivity value at the upper end of the range for brines ($\varepsilon^{\prime }=110$ and tanδ=100, as in Figure 3) and a time delay of the subsurface echo τ = 160 μs, as in [32].

A comparison of loss tangent values in Figure 4 with material properties in Table 1 and plots in Figure 1 reveals that strong subsurface echoes at depths of the order of a kilometer are possible if attenuation in the surface material is similar to that of cold ice with or without a small fraction of dust, and perhaps to that of porous rock, but not if it is that of dense basaltic rocks. Because the estimated depth of the Martian water table is of the order of a few to several kilometers [31], identification of subsurface liquid water outside the polar caps is made challenging by the likely weakening of radar echoes, and in fact it has been predicted that the Martian water table could not be detected at all by MARSIS if its depth is greater than a few kilometers [55].

Because the dielectric constant of materials listed in Table 1 is constant or nearly constant in the MHz to GHz frequency range [46], the values of $P_{ss}/P_{s}$ in Figure 3 can be considered independent from frequency. While the loss tangent of water ice is inversely proportional to frequency [47], the loss tangents reported in Table 1 are frequency-independent. As the term $\exp\left(-2\pi f\tan\delta \;\tau \right)$ in Equation (Equation 4) includes frequency *f*, then attenuation in pure water ice is frequency-independent, while it increases with frequency in all other materials. A decrease of $P_{ss}/P_{s}$ with increasing MARSIS frequencies was already noted in [32]; this phenomenon was interpreted as due to the presence of dust in the ice, making the loss tangent of the Southern Polar Layered Deposits similar to that of low temperature ice/dust mixture shown in Figure 1. This property of dust-contaminated ice was also invoked to explain the absence of basal echoes in radar echoes collected by SHARAD, which is a radar sounder similar to MARSIS aboard NASA’s Mars Reconnaissance Orbiter [39] operating at a central frequency of 20 MHz and transmitting a 10 MHz-bandwidth pulse. Extrapolating the value of $P_{ss}/P_{s}$ at SHARAD frequencies based on the trend observed in MARSIS leads in fact to the prediction that the basal echo will be near or below the detection threshold, as shown in Figure 5.

The discussion above is based on the implicit assumption that the surface layer is homogeneous down to the depth of the interface producing subsurface echoes. This assumption is not verified even in the case shown in Figure 2, the starting point for this analysis, in which the layered structure within the South Polar Layered deposits is clearly visible. Internal layering will result in a loss of energy of the propagating pulse due to multiple reflections, and thus in the weakening of subsurface echoes. Furthermore, if the subsurface interface is topographically rough, as determined through Equation (Equation 8), then the pulse would be scattered in directions that differ from the specular one, thus weakening subsurface echoes even further. Even in the ideal case of a plane parallel geometry and homogeneous media, resonance effects may artificially enhance or depress both the surface and subsurface echoes, leading to measured values of $P_{ss}/P_{s}$ that cannot be explained through the use of Equation (Equation 7) (e.g., [56]).

In the absence of morphological evidence such as that available for terrestrial subglacial lakes, the search for subsurface water through radar sounding is an inverse electromagnetic problem to determine the relative permittivity of the material producing a measured radar echo. This is a complex problem fraught by the lack of knowledge of many parameters such as attenuation within the surface layer and roughness of subsurface interfaces. Several approaches have been proposed over the years (e.g., [52,57,58,59]), but all of them required some ad hoc assumptions that prevented generalization.

## 4. Conclusions and Perspectives

The investigation leading to the discovery of subsurface liquid water on Mars through radar sounding was prompted by the detection of echoes, coming from a depth of about 1.5 km, which were stronger than surface echoes [32]. The cause of this anomalous characteristic is the high relative permittivity of water-bearing materials, resulting in a high reflection coefficient (Equation (Equation 1)). A determining factor in the detectability of such strong echoes is the low attenuation of the MARSIS radar pulse in cold water ice (Figure 1), the main constituent of the Martian polar caps.

The present analysis clarifies that the conditions causing exceptionally strong subsurface echoes occur solely in the Martian polar caps, and that the detection of subsurface water under a predominantly rocky surface layer will require thorough electromagnetic modeling, complicated by the lack of knowledge on many subsurface physical parameters. As signal attenuation in rocks increases with frequency, a future radar operating at frequencies below those of MARSIS could in principle detect deeper and stronger water-related echoes. However, because the maximum plasma frequency of the Martian ionosphere is several hundred kHz even in favorable conditions (i.e., on the night side [60]), it is not possible to probe the subsurface at frequencies much lower than a MHz, which would result in only a modest increase of $P_{ss}/P_{s}$.

The search for strong basal echoes beneath the Martian polar caps is far from being complete, however. As discussed in [32], the small size of strong subsurface reflectors compared to the MARSIS footprint required the use of data that have not been processed on board before being downlinked to Earth, because such processing drastically reduced the radar sampling rate along the ground track. These raw data could be acquired only after a modification of the on-board software and constitute a small fraction of the MARSIS dataset. Coverage of the polar caps in this mode is thus sparse, but it is bound to increase in coming years, so that more bright subsurface reflectors could be potentially discovered in the future.

The Mars Express spacecraft was launched in 2003 and it is thus expected that MARSIS will continue collecting data on the Martian polar caps for no more than a few years. Fortunately, the Chinese mission to Mars to be launched in 2020, Tianwen-1, will carry the Mars Orbiter Subsurface Investigation Radar (MOSIR), which will operate in the 10–15 MHz, 15–20 MHz, and 30–50 MHz frequency ranges. The lowest band is at frequencies that are intermediate between those of MARSIS and those of SHARAD [61]. As shown in Figure 5, such a high frequency range will probably result in weaker subsurface echoes even in the presence of liquid water, but the signal-to-noise ratio of subsurface detection will likely be sufficient to identify anomalously bright subsurface reflectors in comparison to their surroundings.

In spite of its limitations, orbital radar sounding is currently the only technique that allows a global search of subsurface water from orbit. Ground Penetrating Radars (GPR) will be carried by NASA’s Perseverance [62] and China’s Tianwen-1 [63] rovers, to be launched in 2020, as well as by ESA’s Rosalind Franklin rover [64], whose launch has been postponed to 2022. Although capable of a much better resolution, these experiments operate at higher frequencies, and thus cannot penetrate as deep as MARSIS. Furthermore, they lack the mobility needed to achieve large-scale coverage. An alternative electromagnetic method for deep subsurface study is time-domain electromagnetic (TDEM) sounding [65], which works by inducing eddy currents in the subsurface and by measuring the magnetic fields produced by such currents. This technique allows the determination of subsurface conductivity, which increases by orders of magnitude in the presence of saline water, and can achieve deeper penetration than GPR at the cost of lower resolution. However, because the size of the loop used to induce ground currents must be comparable to the depth of probing and because the loop needs to be close to the medium in which currents are to be induced, this method is not suitable for orbiting platforms.

The exploration of the Martian subsurface is critical in the search for life on Mars [66]. Stable bodies of subsurface water are considered among the most promising potential habitats existing on today’s Mars (although isolation from the surface could prevent the actual presence of life [67]), and detecting them remains one of the prime goals of Martian exploration. In spite of a long-sought initial success, much work is still to be done before the search for subsurface water can be considered concluded.

## Figures and Tables

**Figure 1 life-10-00120-f001:**
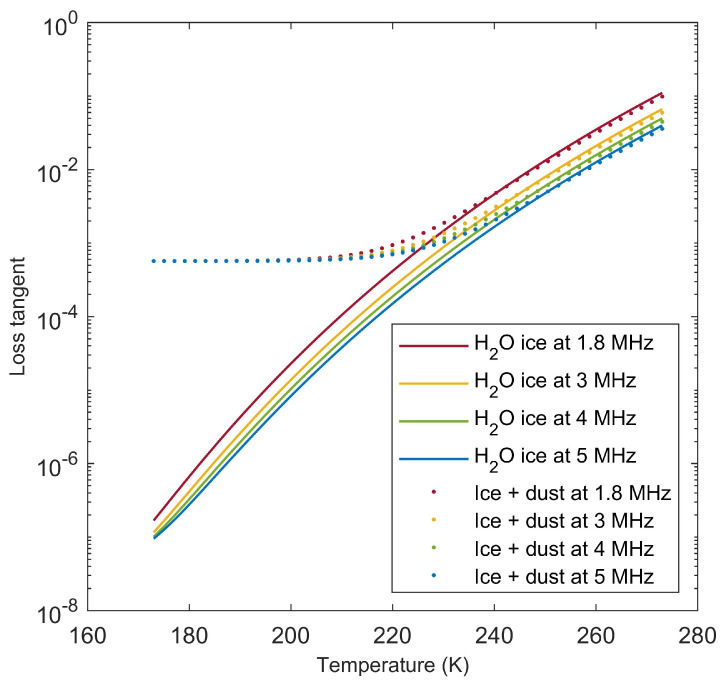
Loss tangent of pure water ice and of an ice/dust mixture with a volumetric fraction of dust equal to 0.1 as a function of temperature, for the four Mars Orbiter Subsurface Investigation Radar (MOSIR) operating frequencies. The loss tangent of water ice is computed according to formulas presented in [47], while the permittivity of volcanic rock in Table 1 has been used to represent that of dust. The effective permittivity of the ice/dust mixture has been obtained through Equation (Equation 6).

**Figure 2 life-10-00120-f002:**
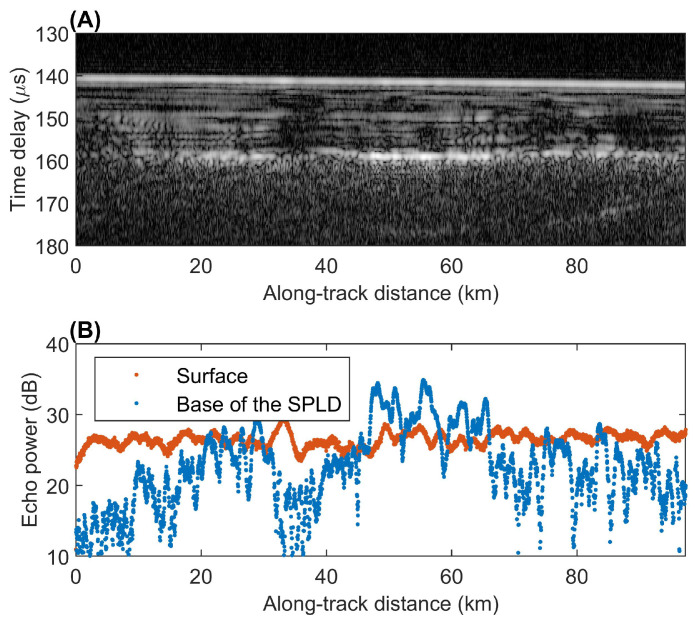
(**A**) Radargram for MARSIS orbit 10737. A radargram is a bi-dimensional color-coded section made of a sequence of echoes in which the horizontal axis is the distance along the ground track of the spacecraft, the vertical axis represents the two-way travel time of the echo (from a reference altitude of 25 km above the reference datum), and brightness is a function of echo power. The continuous bright line in the topmost part of the radargram is the echo from the surface interface, whereas the bottom reflector at about 160 μs corresponds to the interface between the Southern Polar Layered Deposits (SPLD) and the bedrock. Strong basal reflections can be seen at some locations, where the basal interface is also planar and parallel to the surface. (**B**) Plot of surface and basal echo power for the radargram in (**A**). Red dots mark surface echo power values, while blue ones mark subsurface echo power. The horizontal scale is along-track distance, as in (**A**), while the vertical scale reports uncalibrated power in decibels (dB). The basal echo between 45 km and 65 km along track is stronger than the surface echo even after attenuation within the SPLD (adapted from [32]).

**Figure 3 life-10-00120-f003:**
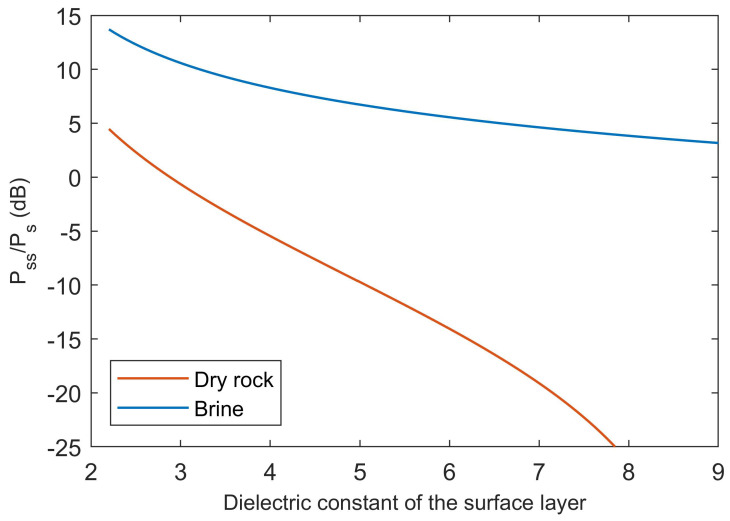
Values of $P_{ss}/P_{s}$ computed according to Equation (Equation 7) for a bedrock consisting of dry volcanic rock and a body of subglacial brine, by varying the dielectric constant value of the surface layer between that of CO${}_{2}$ ice and that of dense volcanic rock, and by assuming that there is no signal attenuation due to dielectric losses in the surface material (see text for details).

**Figure 4 life-10-00120-f004:**
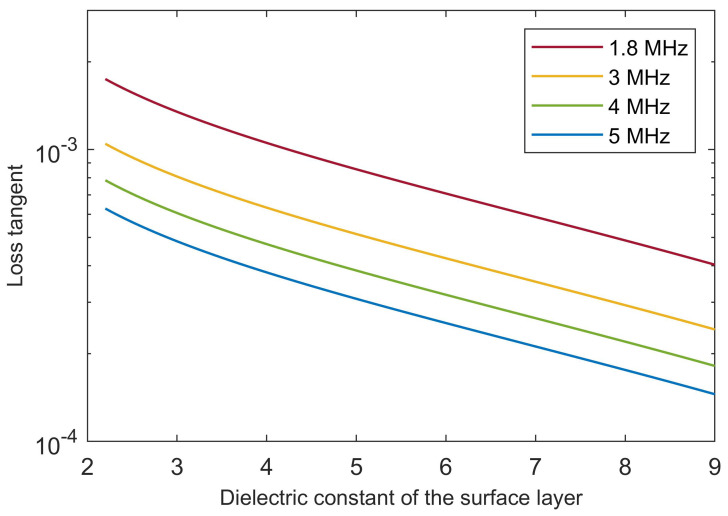
Values of surface material loss tangent that result in $P_{ss}/P_{s}=1$ according to Equation (Equation 7), for a basal relative permittivity value at the upper end of the range for brines and a time delay of the subsurface echo of 160 μs, as in [32] (see text for details).

**Figure 5 life-10-00120-f005:**
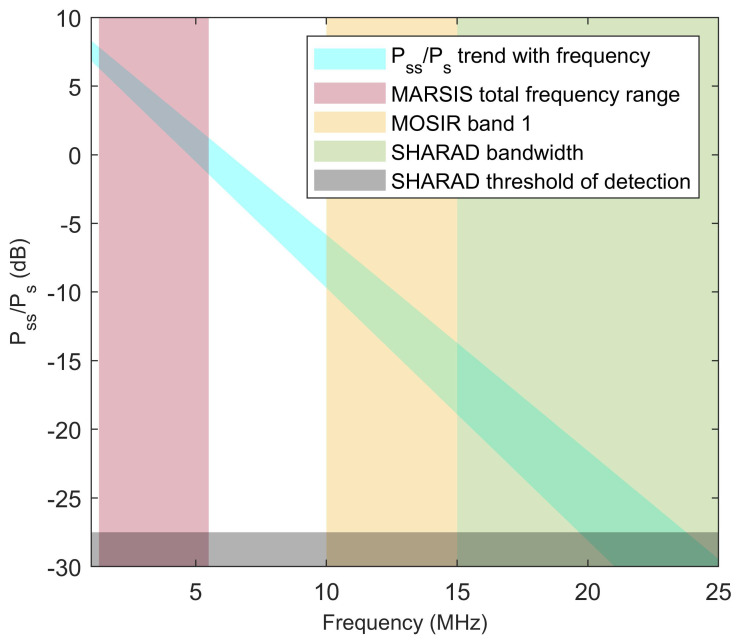
Estimate of the ratio of subsurface to surface echo power over the bright reflector in [32] as a function of frequency, extrapolated from MARSIS data. The light blue diagonal strip represents the area of the best fit to the data extending to the 90% confidence level. The colored rectangles highlight the operation bands of different radar instruments.

**Table 1 life-10-00120-t001:** Values of the complex relative permittivity of materials present on the Martian surface in the MHz frequency range.

Material	Dielectric Constant $\varepsilon^{\prime }$	Loss Tangent tanδ	Source
Volcanic rocks	4–9	$10^{-3}$–$10^{-2}$	[45,46]
H${}_{2}$O ice	3.1	$10^{-7}$–$10^{-1}$	[47]
CO${}_{2}$ ice	2.2	$4\times 10^{-3}$	[48]
Water	≈80	$\approx 10^{-3}$	[46]
Brine	80–110	10–100	[46]

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
