# Peer review of "The Global Search for Liquid Water on Mars from Orbit: Current and Future Perspectives"

_life, 2020, doi:10.3390/life10080120_

Round 1
Reviewer 1 Report
An interesting and informative review of the topic. The description of the physics is perhaps a little deep for an astrobiology theme, but nevertheless the review's level is well pitched.
I note no major errors or corrections required.
Author Response
We are grateful for the reviewer's appreciation. Following comments by Reviewer 2, we have tried to clarify some of the more technical aspects of the discussion.
Reviewer 2 Report
The authors have written a compact review of the methodology and evidence for identifying subsurface bodies of water on Mars. It is clearly written and the results are laid out systematically. There are however a number of points the authors should address.
(1) The authors should clarify what they mean by lines 36-38. Specifically, why and how were the ice accumulation deposits different than the current ones.
(2) Ref. [13] used a simple analytical model to estimate the amount of water/atmosphere lost over Mars' history. A more accurate result was obtained the same year, which should be cited here:
C. Dong, Y. Lee, Y. Ma, et al., Modeling Martian atmospheric losses over time: implications for exoplanetary climate evolution and habitability, Astrophys. J. Lett., 859, L14 (2018)
(3) In lines 84-86, the authors briefly mention that other mechanisms could cause the same geological features, but state that they were formed by debris flows involving water. This point should be explained: why is this hypothesis favoured over others?
(4) In the last paragraph of Section 2, the authors should mention any competing hypotheses to explain the same phenomenon, i.e., the change in relative permittivity measured at 1.5 km depth.
(5) Equation (3) is similar to the power collected over a solid angle Omega = theta^2, where theta = G lambda/(8 pi H), times the reflectivity.
(6) The authors say in lines 167-168, the loss tangent of water ice is orders of magnitude smaller than other substances. This is not correct: water ice can have loss tangent of ~0.1, and indeed this is present in the Table. Hence, this statement should be more carefully rewritten.
(7) In plotting Figures 3 and 4, there are many free parameters that are set equal to some fiducial values. Examples include T_s, R_s, and R_ss. Perhaps the authors can add a brief Appendix or just include these values in the text. One of the important points about science is reproducibility, and adding these values will make it easier for the readers instead of going back and trying to consult a lot of other references.
(8) At the conclusion of the paper, the authors state that "Stable bodies of subsurface water are considered the only potential habitat existing on today’s Mars..." This statement is too strong given the number of uncertainties that still remain about our understanding of biochemistry, extremophiles, Martian geology, etc. A good reference to consult, and even cite here, is the following, which presents a more comprehensive overview:
C. Cockell, Trajectories of martian habitability, Astrobiology, 14, 182 (2014)
Author Response
We wish to thank Reviewer 2 for constructive comments that have allowed us to improve some unclear parts of the manuscript. A point-by-point reply to individual comments is reported below.
(1) The authors should clarify what they mean by lines 36-38. Specifically, why and how were the ice accumulation deposits different than the current ones.
Authors' reply - We have extended the sentence as follows:
The Basal Unit (BU) is a deposit consisting of water ice and lithic fines, lying stratigraphically beneath the North Polar Layered Deposits. It consists of two geologic units, namely the Rupes Tenuis unit at the bottom, and the Boreum Cavi unit on top, both of Amazonian age. The Boreum Cavi unit appears to consist predominantly of sandy material [2]. The Dorsa Argentea Formation (DAF) is a vast Hesperian-aged unit surrounding and partially underlying the South Polar Layered Deposits. Volcanic activity, debris flows, aeolian deposition and glacial activity have been proposed as formation mechanisms [2], but the hypothesis that the DAF is the remnant of a large ice sheet [7] seems to be better supported by evidence.
(2) Ref. [13] used a simple analytical model to estimate the amount of water/atmosphere lost over Mars' history. A more accurate result was obtained the same year, which should be cited here:
C. Dong, Y. Lee, Y. Ma, et al., Modeling Martian atmospheric losses over time: implications for exoplanetary climate evolution and habitability, Astrophys. J. Lett., 859, L14 (2018)
Authors' reply - We have added a reference to the work of Dong et al. (2018).
(3) In lines 84-86, the authors briefly mention that other mechanisms could cause the same geological features, but state that they were formed by debris flows involving water. This point should be explained: why is this hypothesis favoured over others?
Authors' reply - We have modified and extended the sentence as follows:
Terrestrial formation mechanisms that have been considered potential analogs for Martian gullies include pyroclastic flows and dry snow avalanches (as examples of natural dry granular flows), and fluvial flows, debris flows and slushflows as processes involving the presence of liquid water. As discussed in [23], morphological evidence and laboratory experiments seem to point to liquid-water debris flows resulting from surface melting as the most plausible formation mechanism for gullies.
(4) In the last paragraph of Section 2, the authors should mention any competing hypotheses to explain the same phenomenon, i.e., the change in relative permittivity measured at 1.5 km depth.
Authors' reply - We have added the following text to the paragraph:
Alternative mechanisms producing strong basal echoes are the presence of a CO2 ice layer at the top or the bottom of the SPLD, or a very low temperature of the H2O ice throughout the SPLD, enhancing basal echo power compared to surface reflections. However, such phenomena either require very specific physical conditions, or they do not cause sufficiently strong basal reflections.
(5) Equation (3) is similar to the power collected over a solid angle Omega = theta^2, where theta = G lambda/(8 pi H), times the reflectivity.
Authors' reply - We have added the following explanation of the meaning of the geometric term in Equation (3):
The geometric term (l/(8pH))2 represents the geometric loss due to the spherical expansion of the wavefront (both in transmission and after reflection) multiplied by the squared area of the Fresnel circle producing the specular reflection.
(6) The authors say in lines 167-168, the loss tangent of water ice is orders of magnitude smaller than other substances. This is not correct: water ice can have loss tangent of ~0.1, and indeed this is present in the Table. Hence, this statement should be more carefully rewritten.
Authors' reply - The statement has been made before discussing Figure 1, and it is thus confusing. The paragraph immediately following Table 1 has thus been reworded as follows:
Most materials listed in Table 1 do not exhibit a strong dependence of their relative permittivity on temperature in the range expected for the Martian surface. The loss tangent of water ice, however, increases by orders of magnitude with temperature [47], drastically affecting attenuation of the radar wave (see Equation (4)). The loss tangent can be computed as a function of temperature using formulas presented in [47], and is shown in Figure 1 below. It can be seen that cold, pure water ice has a loss tangent that is orders of magnitude below that of other substances, resulting in very little attenuation, thus extremely transparent to radar waves. As ice temperature approaches the melting point, its tan d becomes higher than that of rock, thus strongly limiting penetration of the radar signal.
(7) In plotting Figures 3 and 4, there are many free parameters that are set equal to some fiducial values. Examples include T_s, R_s, and R_ss. Perhaps the authors can add a brief Appendix or just include these values in the text. One of the important points about science is reproducibility, and adding these values will make it easier for the readers instead of going back and trying to consult a lot of other references.
Authors' reply- We acknowledge we have not provided sufficient information for the reader to reproduce computations. The paragraph immediately preceeding Figure 3 and the figure caption have thus been modified as follows:
The relative permittivities of surface and subsurface materials have to be defined to compute values of Pss/Ps through Equation (7). The dielectric constant of the surface layer is varied between the lowest value for dry materials reported in Table 1, corresponding to 2.2 for CO2 ice, to a value of 9 matching dense volcanic rocks, to explore the effect of surface layer composition on the strength of subsurface echoes. To simplify the analysis, and as a way to maximize Pss/Ps by neglecting dielectric losses, the loss tangent of the surface layer is assumed to be negligible. To compare the most favorable cases for the occurrence of strong subsurface reflections by dry and water-bearing materials, the relative permittivity of the dry bedrock has been assumed to be the highest in the range of values for dry volcanic rocks in Table 1, that is #0 = 9 and tan d = 10?2, while the relative permittivity of the liquid water body has been taken to be the highest reported for brines in Table 1, namely #0 = 110 and tan d = 100. Brines are considered to be more plausible than liquid water as the source of strong basal echoes because temperatures at the base of the SPLD are expected to be well below the freezing point of water (see discussion in [32]). Once the relative permittivities have been defined, the terms Rss, Rs and Ts in Equation (7) can be computed by means of Equations (1) and (2).
Figure 3. Values of Pss/Ps computed according to Equation (7) for a bedrock consisting of dry volcanic rock and a body of subglacial brine, by varying the dielectric constant value of the surface layer between that of CO2 ice and that of dense volcanic rock, and by assuming that there is no signal attenuation due to dielectric losses in the surface material (see text for details).
(8) At the conclusion of the paper, the authors state that "Stable bodies of subsurface water are considered the only potential habitat existing on today’s Mars..." This statement is too strong given the number of uncertainties that still remain about our understanding of biochemistry, extremophiles, Martian geology, etc. A good reference to consult, and even cite here, is the following, which presents a more comprehensive overview:
C. Cockell, Trajectories of martian habitability, Astrobiology, 14, 182 (2014)
Authors' reply - We have modified the statement as follows:
Stable bodies of subsurface water are considered among the most promising potential habitats existing on today’s Mars (although isolation from the surface could prevent the actual presence of life [67]), and detecting them remains one of the prime goals of Martian exploration.
Finally, we have made some minor additions and corrections to the text. In particular, at the end of Section 2 we added and discussed a recent paper that we felt relevant. We also corrected the value for the relative permittivity of brines in Table 1 and applied it to the analysis of subsurface water detectability, modifying some parts of the text for consistency but without affecting conclusions. The revised paper highlighting text modifications is attached below.

Round 2
Reviewer 2 Report
The authors have done a thorough job of implementing the changes. I believe that this review constitutes a good addition to the journal.